# Outcomes for Patients Receiving Multi-Chamber Bags for the Delivery of Parenteral Nutrition: A Systematic Review

**DOI:** 10.3390/nu16223964

**Published:** 2024-11-20

**Authors:** Debra Jones, Karen Allsopp, Anne Marie Sowerbutts, Simon Lal, Kirstine Farrer, Simon Harrison, Sorrel Burden

**Affiliations:** 1School of Health Sciences, University of Manchester, Oxford Road, Manchester M13 9PL, UK; karen.allsopp@manchester.ac.uk (K.A.); annemarie.sowerbutts@manchester.ac.uk (A.M.S.); sorrel.burden@manchester.ac.uk (S.B.); 2Salford Care Organisation, Northern Care Alliance NHS Trust, Stott Lane, Salford M6 8HD, UK; simon.lal@nca.nhs.uk (S.L.); kirstine.farrer@nca.nhs.uk (K.F.); simon.harrison@boltonft.nhs.uk (S.H.)

**Keywords:** intestinal failure, parenteral nutrition, home parenteral nutrition, parenteral nutrition manufacture, standardised multi-chamber bags, individually compounded bags, nutrients, fluids

## Abstract

Background: Parenteral nutrition (PN) is required by people with intestinal failure and can be delivered as multi-chambered bags (MCBs) or individually compounded (COM) bags. This systematic review aimed to examine the evidence base for clinical outcomes and/or quality of life (QoL) in adults receiving PN as MCBs compared to COMs in hospital and community settings. Methods: A systematic database search was conducted between January 2015 and May 2024. Studies assessing adults in receipt of MCBs were included. Quality was assessed using Joanna Briggs appraisal tools. A narrative synthesis was performed due to study heterogeneity. PROSPERO: CRD42022352806. Results: Ten studies including 87,727 adults were included, with 20,192 receiving PN from MCBs and 67,535 from COMs. Eight studies reported on PN given in hospital and two in the home. Five hospital-based and one home-based study reported that MCBs were well tolerated and provided adequate nutrition. Three hospital-based studies reported that MCBs had lower post-operative infections and a lower mean risk of catheter-related bloodstream infections (CRBSIs). Two home-based studies reported no difference in CRBSI. Five hospital-based studies reported no difference between groups in length of hospital stay. Three hospital-based studies reported the cost to be lower for MCBs than COMs, and no studies reported QoL. Conclusions: The studies included show that MCBs provided in hospital are safe and non-inferior to COMs and may be more cost-effective. There were few high-quality studies and no data on QoL; therefore, further work is required to improve the certainty of the evidence and to establish the level of QoL when using MCBs.

## 1. Introduction

Parenteral nutrition (PN) is the intravenous provision of macronutrients provided in the form of a solution of amino acids, glucose, and lipids, with electrolytes, vitamins, trace elements, and fluid [1]. PN is essential for survival in people who have intestinal failure (IF), which is “the reduction of gut function below the minimum necessary for the absorption of macronutrients and/or water and electrolytes, such that intravenous supplementation is required to maintain health and/or growth” [2,3]. IF can be a relatively short-term condition, whereby patients receive PN for a limited period of time in hospital [4], or a chronic condition, where patients require long-term, sometimes lifelong PN at home (termed home PN) [5]. PN is primarily available in compounded bags (COMs), which are tailored to the individual patient’s specific daily nutritional and fluid requirements, or multi-chamber bags (MCBs), which are commercially available in different standardised formulations and volumes.

MCBs are terminally sterilised products with validated stability limits that do not require refrigeration [6]. MCBs are available as either a two-chamber bag, containing glucose/dextrose and amino acids, or a three-chamber bag, containing glucose/dextrose, amino acids, and intravenous lipid emulsions [7]. However, unlike bespoke, tailor-made COMs [6], MCBs do not contain vitamins and trace elements, so that the latter must be given orally, intravenously, or as a separate infusion [8]. Supplemented MCBs can have intravenous vitamins and minerals added to the bag, but this reduces the shelf life of the product, although it has been suggested that increasing the range of MCBs available would attenuate this issue [7]. However, operationally, many countries inject micronutrients into the infusion [9]. In addition, MCBs tend to contain relatively conservative amounts of electrolytes, such that not all MCBs may meet the patient’s exact needs. As a result, patients may require additional infusions of vitamins and/or electrolytes compared to COMs in order to meet their exact requirements [8,10]. The additional infusions can mean that patients require increased manipulation of the central venous catheter to infuse the PN, which in turn may increase the risk of catheter-related bloodstream infections [10,11].

Contrary to this, others have shown that the use of MCBs in comparison to COMs for HPN patients is not associated with an increased risk of catheter-related bloodstream infections (CRBSIs) [12,13,14]. In addition, the workforce and aseptic facility requirements for COM preparation may enhance their costs compared to MCBs, although this may be dependent on the number of patients being treated per day [15]. It is also important to note that in the UK, MCBs, which are licensed medicinal products, are advised to be used ahead of an unlicensed fully COM (bespoke) regimen, unless there is a clear clinical need [8]. Furthermore, the type of regimen may impact on a patient’s quality of life (QoL), particularly if the patient requires long-term PN at home [16]. There appears to be little evidence in the literature relating to differences in QoL between COMs and MCBs, although previous work has shown that home PN has a detrimental impact on QoL due to the related restrictions on travel and socialising [17]. QoL has been reported as one of the top research priorities of patients receiving PN, their family members, and healthcare professionals involved in HPN provision [18]. Despite this, a survey asking about healthcare professionals’ opinions on QoL in the care of patients receiving HPN found that the majority of healthcare professionals perceived QoL assessment as useful and important but very few embedded it into clinical practice [19].

The exact risks and benefits of MCBs vs. COMs have not been recently comprehensively evaluated, either in the hospital or home settings, since Alfonso and colleagues performed a systematic review comparing MCBs vs. other delivery systems in hospitalised patients. This review concluded that the use of MCBs may be associated with a lower risk of bloodstream infections compared to other delivery systems albeit the evidence was assessed as low quality overall [20]. Indeed, the European Society of Parenteral and Enteral Nutrition (ESPEN) guidelines stated HPN solutions can be prepared either by individual compounding or by “adapted commercial MCBs”. They reiterate that there is a “paucity of data in the HPN setting” on the use of MCBs compared to COM PN [1]. The aim of this systematic review is to update the review conducted by Alfonso and colleagues by examining evidence published since 2014 on clinical outcomes and QoL in adults receiving PN in the form of MCBs compared to individually COM PN in both the hospital and home settings.

## 2. Materials and Methods

### 2.1. Reporting Guidelines and Protocol Registration

This review followed the PRISMA 2020 guidance for reporting systematic reviews, and the PRISMA 2020 checklist for this review can be found in Appendix A. In accordance with the guidelines, our systematic review protocol was registered with the International Prospective Register of Systematic Reviews (PROSPERO) on 11 August 2022. (Registration number: CRD42022352806. Available from: https://www.crd.york.ac.uk/PROSPERO/display_record.php?RecordID=352806 (accessed on 15 November 2024).

Differences from protocol: This review originally aimed to assess the outcomes with MCBs in patients with intestinal failure. However, the literature search revealed that the term “intestinal failure” was rarely used in international literature beyond the UK. Therefore, we modified the criteria to include all studies of MCBs regardless of the indication for which PN was being provided.

### 2.2. Eligibility Criteria

Population: We included randomised controlled trials (RCTs), controlled (non-randomised) clinical trials or cluster trials, prospective and retrospective comparative cohort studies, cross-sectional studies, qualitative studies using grounded theory, phenomenology thematic analysis or case studies, and case–control studies (with at least 5 or more cases) that examined adult human patients receiving PN. We excluded studies or data sets within studies where patients were receiving intravenous fluids and electrolytes only with limited or no macronutrients. Studies with individuals all under the age of 18 were excluded. If the age of the subjects included was 18 or over for more than 80% of participants, these studies were included. Editorials, narrative reviews, and abstracts without full-text articles were excluded.

Interventions: Interventions or exposures that included the provision of standardised PN delivered using MCBs.

Comparators: Comparators included PN delivered using COMs.

Outcomes: The outcomes included the following: proportion of nutritional requirements met for macro- and micronutrients; maintenance or improvement of nutritional status; patient experience; and QoL and clinical outcomes, including the number and type of complications and mortality. Additional outcomes included the production time and cost of PN. Outcomes were extracted as reported in all included studies, in all data forms (e.g., dichotomous, continuous), using a pre-defined data extraction form.

Settings: Studies taking place in both the home and hospital settings were included.

Years, language, and publication status: Articles published from November 2014 up to the present day were included. Articles reported in any language and describing original research published in peer-reviewed scientific journals were included. A summary of the inclusion and exclusion criteria can be viewed in Appendix A.

### 2.3. Information Sources

Searches were conducted up to 2 May 2024. Searches were run in six health and social care databases, with results limited to humans only; the databases included MEDLINE (Ovid interface, U.S. National Library of Medicine), EMBASE (Ovid interface, Elsevier), CINAHL (EBSCOhost interface, EBSCO Publishing), Cochrane Central Register of Controlled Trials (Wiley interface, latest issue, John Wiley & Sons, Inc.), PsycInfo (Ovid interface, American Psychological Association), and Web of Science (Clarivate Analytics interface, Clarivate). To ensure literature saturation, PROSPERO was searched for any ongoing or completed reviews, and the reference and citation lists of all the included studies were scanned and manually screened to identify any other relevant articles. All languages were included, and a translation service (Google translate) was used to translate non-English articles.

### 2.4. Search Strategy

Literature search strategies were developed using medical subject headings (MeSHs) and text words related to the method of delivery of PN. A preliminary review of a sample of the literature revealed that there is a lack of standardised nomenclature for PN bag types, and descriptions of bags varied widely. The preliminary review was used to gather the varied terminology used for PN bag types across the international literature; see Appendix A. The wording and terminology in Appendix A was used to further develop the terms used in the search strategy. Search strategies were run simultaneously for six databases and the results deduplicated using each of the platforms’ deduplication tools. The full search strategies for all databases can be found in Appendix A.

### 2.5. Selection Process

The literature search results were uploaded to Endnote (v20.3, Clarivate) for organisation and then Rayyan (software as a service) [21] for screening. Any remaining duplicates were identified and removed by both the Endnote and Rayyan software; any further remaining duplicates were removed manually. Two reviewers (KA, AMS or DJ) independently screened the titles and abstracts against the inclusion criteria. The full text was obtained for any studies that met the inclusion criteria or where there was any uncertainty. The review authors screened the full-text reports and decided whether they met the inclusion criteria. Disagreements between the reviewers were resolved through discussion and arbitration with a third author (KA, AMS, or DJ) if necessary. The reasons for excluding trials were noted and ordered.

### 2.6. Data Collection

Data were extracted from the included studies using a standardised data extraction form, developed a priori. Data extraction was carried out by one reviewer (KA, AMS, or DJ) and then verified by another (KA, AMS, or DJ) to reduce bias and errors. Any disagreements were resolved through discussion and a third author (KA, AMS, or DJ) consulted for arbitration when necessary.

### 2.7. Data Outcomes

#### 2.7.1. Outcomes

The main outcomes of interest were whether participants met their nutritional requirements and maintained their nutritional status, patient experience, and the impact on clinical outcomes. To measure this, the eligible outcomes were broadly categorised as follows:If participants met their nutritional requirements from PN, plus any additional feeding and oral intake;Participants’ nutritional status and if this was maintained/decreased/increased;Patient experience: Any qualitative reports of experience or QoL questionnaires;Complications: Overall infections/sepsis, metabolic disturbances, mortality, length of hospital stay, readmission, PN modification.

There were no restrictions placed on the number of time points at which the outcomes were measured, nor were there restrictions on the length of follow-up.

In addition, we recorded the associated production and cost implications by extracting the following data variables:Cost of PN. (When reported in different currencies, the cost was extracted into the extraction table in its original currency and then later converted to Euros, for ease of comparison, using the online XE currency converter [22]. At the time of conversion, 1 US dollar = 0.918 Euros and 1 Chinese Yuan Renminbi = 0.126 Euros);Preparation time;Number of modifications to PN prescription.

#### 2.7.2. Other Variables

Other data variables were as follows:Background information: Author, year, journal, language, title, funding, country;Study characteristics: Study design, inclusion/exclusion criteria, setting, sample size;Participant characteristics: Mean age, gender, ethnicity, medical/surgical condition or reason for PN, comorbidities;Exposure characteristics: MCBs/COMs, nutritional content of the bags, duration of PN and duration of follow-up, any additional feeding or oral intake.

### 2.8. Study Risk of Bias Assessment

The quality of the studies was assessed using the Joanna Briggs critical appraisal tools [23]. Joanne Briggs tools consider the methodological quality of studies and then determine the extent to which each study has addressed the possibility of bias in its design, conduct, and analysis. The Joanna Briggs suite includes several tools, and we selected the appropriate tool to appraise each study according to its design. Each checklist uses a series of criteria that can be scored as being “met”, “not met”, or “unclear” and in some instances as “not applicable”. If there was insufficient detail reported in the study, the quality of the study was assessed as “unclear” and the original study investigators were contacted for more information. These judgements were made independently by two review authors (AMS, KA or DJ). Disagreements were resolved first by discussion and then by consulting a third author for arbitration (AMS, KA, or DJ).

### 2.9. Effect Measures and Synthesis of Data

We planned to analyse the success of the outcomes (i.e., maintenance of nutritional status or meeting nutritional requirements) by calculating the risk ratio (RR). However, for quantitative studies, it was not possible to pool the data statistically using meta-analysis due to the low number of studies and the heterogeneity in reporting, and so a narrative synthesis of the results was conducted. We therefore could not conduct risk of bias or certainty of evidence assessments for outcomes due to the presentation of the data and missing values.

## 3. Results

### 3.1. Study Selection

We found a total of 468 records from searching databases, and after removing duplicates, 330 records were screened. From these, 40 full-text documents were reviewed, and 10 studies were included [12,24,25,26,27,28,29,30,31,32]. The PRISMA flow diagram can be viewed in Figure 1. Of the full-text documents reviewed (*n* = 40), 30 were excluded, and the reasons are listed on the PRISMA flow diagram. Of the 10 included papers, eight were published in the English language [12,24,25,26,27,28,31,32] and two in Chinese [29,30]. These were translated using the Google Translate app, and the Chinese language translations were checked and amended for accuracy by a native Chinese-speaking national.

### 3.2. Study Characteristics

#### 3.2.1. Population and Study Design

Across the 10 studies, there were 87,727 patients, and of these 48,606 (55.4%) were female. Eight studies (*n* = 1670) reported mean ages (SD) according to PN bag type, ranging from 53.0 (17.1) to 63 (15.41) years in the MCBs group and 56.3 (11.7) to 66 (13.21) years in the COM bag group [12,25,26,27,29,30,31,32]. One study reported the mean age (SD) for the total sample as 62.5 (15.4) years [28], and another study reported the numbers of participants within age ranges from 18 up to 85+ years [24]. The included studies were conducted across six countries; five studies were conducted in China [27,29,30,31,32] and one study each in America [12], the United Kingdom [25], Spain [25], Singapore [26], and Korea [28]. In terms of study design, five were cohort studies [12,25,26,30,32]; three were randomised controlled trials [27,29,31]; and two were cross-sectional studies [24,28]. Further details of the studies and population characteristics can be found in Table 1.

#### 3.2.2. Exposure Characteristics

All 10 studies compared patients receiving PN from MCBs to patients receiving COM PN. Across the 10 studies, there were 20,192 patients receiving PN from MCBs and 67,535 patients receiving COM PN. Eight of the 10 studies reported on PN received in hospital settings [24,26,27,28,29,30,31,32], with two reporting on PN in the home setting [12,25]. The main indication for receiving PN varied across the studies; seven studies stated that patients had gastrointestinal diseases or complications or had undergone surgery [25,26,27,29,30,31,32], one stated that patients had a nutritional deficiency without giving a specific cause [24], another specified that patients had chronic IF primarily due to the consequences of cancer [12], and the final study listed a number of reasons, including neoplasms, surgery, diseases of the digestive or respiratory or circulatory systems, and diseases of the blood or blood-forming organs [28].

The mean duration of PN for seven of the eight hospital-based studies ranged from 13–15 h to 17.8 days [24,26,27,28,29,31,32], with the remaining study not reporting the duration [30]. For the two studies set at home, one stated a mean duration ranging from 99 to 411 days [12] and the other a median duration of 64 days [25].

The nutritional content of the PN bags varied and was described in 8 of the 10 studies [25,26,27,28,29,30,31,32]; see Table 2 for details. Seven studies, including the two based in the home setting, did not state whether patients were concurrently having oral food intake or tube (enteral) feeding [12,24,25,26,28,30,32]. One study, which had a mean PN duration of 8.2 to 8.4 days, stated nil by mouth from days 0 to 5, and from day 6 onwards liquid oral or enteral nutrition could be added to the treatment [27]. Another study stated that oral and/or enteral nutrition provided up to 20% of the daily energy intake [31], and one study stated that patients were excluded if they were receiving other nutritional therapy [29]. Further details of the exposure characteristics can be found in Table 2.

### 3.3. Risk of Bias in Studies

The Joanna Briggs critical appraisal tool [23] was used to assess the risk of bias in the included studies. The detailed appraisal is displayed in Appendix A. In terms of overall quality, three studies were high quality [12,24,25], five studies medium quality [27,28,29,31,32] and two studies were low quality [26,30]. Some concerns were raised concerning confounding in four of the assessed studies [26,28,30,32], where there was either a lack of identification or lack of control for confounders; however, the uncontrolled confounding within these studies was not considered to be substantial. Concerns were raised for the measurement of the exposure in one study [30], where there was a lack of clarity as to the nutritional content of the PN bags. Concerns were also raised over the measurement of the outcomes in five of the assessed studies, where either the follow-up time was thought to be inadequate [26]; key information was missing, such as details on how the outcomes were measured [27,29,32]; or the outcomes were recorded as frequencies and occurrences and were not compared back to the type of PN administration [28]. In addition, despite there being no initial concerns raised over the selection of participants into any of the studies, the quality of six studies may be further downgraded as they were all conducted retrospectively and, therefore, would have been inadvertently subjected to selection bias [12,24,26,28,30,32]. If selection bias is considered, then the number of studies considered to be high quality reduces to one [25] and the number of studies considered to be low quality increases to four [26,28,30,32].

### 3.4. Results of Individual Studies

#### 3.4.1. Nutrition, Weight, and Patient Experience

Six studies out of ten mentioned or recorded data relating to nutritional requirements and body mass index (BMI) or body weight [25,26,27,29,31,32]; five of these were set in the hospital [26,27,29,31,32] and one in the home [25]. Of the five set in the hospital, four stated that MCBs were effective in preserving nutritional adequacy compared to COM PN, with one study reporting no difference in BMI [25,32]; one reported no differences in energy, protein, or electrolyte intake [26], and one reported comparable trends of prealbumin and post-operative nutritional index [29], and another found no difference in the time to achieve adequate enteral nutrition following five days of PN (median time: MCBs, olive group 2.0 days versus COM, soybean group 2.0 days; log rank *p* = 0.786) [27]. Two of the hospital studies reported that MCBs improved certain nutritional factors, both stating that prealbumin increased in the MCBs group compared to COMs [27,31]. One stated that anabolic and catabolic status was improved, referring to the measurement of acute-phase response protein, prealbumin (least square geometric mean ratio 1.12 [95% CI 1.06 to 1.19], *p* = 0.0002), and oleic acid (least square geometric mean ratio 1.22 95% CI 1.09 to 1.37, *p* = 0.0006) after 5 days of treatment [27].

The one study set in the home agreed that MCBs were effective in preserving nutritional adequacy compared to COM PN by reporting no difference in BMI [25].

However, none of the studies, set in the home or hospital, reported on patients’ experiences or QoL. Four studies did not report any data on nutrition, weight, or patient experience [12,24,28,30]. Further details on nutritional status outcomes can be found in Table 3.

#### 3.4.2. Clinical Outcomes

All 10 hospital and home studies reported on clinical outcomes, apart from one hospital-based study [26], which reported that the PN was modified and participants receiving either MCBs or COMs received additional potassium, phosphate, and magnesium [24].

Five of the ten included studies reported risks or the occurrence of infections during treatment. Three out of the five were hospital-based and reported that MCB groups had lower numbers of post-operative infections (MCBs: 13 and COMs: 33, *p* = 0.003) [30] and overall infections (MCBs: 8 and COMs: 24, *p* < 0.01) [27], and there was a lower adjusted mean risk of CRBSIs in the MCB-only group (MCBs: 2.1%, 95% CI 2.0 to 2.2 and COMs: 6.8%, 95% CI 6.7 to 6.9), although the MCB with additions group was found to have a similar risk to COMs (MCB with additions: 7.0%, 95% CI 6.8 to 7.2) [24]. The other two studies were set in the home and reported no difference in catheter-related bloodstream infections per catheter days; MCBs: 0.21/1000 days and COMs: 0.28/1000 days, *p* > 0.05 [25], MCBs: 0.51/1000 days and COMs: 0.39/1000 days, incident ratio 1.29 (95% CI 0.17 to 9.65) [12].

Five studies reported on metabolic disturbances. Four of these were set in the hospital and stated that either clinical chemistry measures were within reference ranges [27] or post-operative inflammatory markers were similar between PN groups [29], or they reported no differences apart from higher increases in bilirubin levels in MCBs compared to COMs, with an increase of 6.74 (SD 12.18) and decrease of 0.77 (SD 4.77), respectively [32], and significant differences in prothrombin time, alkaline phosphatase (ALP), triglycerides, cholesterol, potassium, chloride, and phosphate, although the results were not clinically important [31]. The one home-based study reported that the occurrence of metabolic complications was low to non-existent (MCBs: 0, COMs: 1) [25].

Four studies reported mortality. Three were set in the hospital, with one reporting very low numbers of deaths (MCBs: 1 and COMs: 4) [27], and two reported no deaths in either PN group [30,31]. The one home-based study stated similar median survival days between groups (MCBs: 88, 95% CI 43 to 133 and COMs: 98, 95% CI 49 to 147; *p* = 0.913) [25].

For length of hospital stay, there were five hospital-based studies, which reported no differences between groups in length of hospital stay (MCBs: 12.4 to 19 days; COMs: 12.3 to 24 days) [24,27,30,31,32]. One study reported on readmission and stated that comparing the MCB groups, the MCBs-only group had a lower risk of having 30-day or 90-day all-cause readmission, whereas the MCB with additions group had a higher risk of 30-day or 90-day all-cause readmission (30 day—MCB only: 18.5%; COMs: 20.5%; MCBs with additions: 22.0%) (90 day—MCB only: 26.8%; COMs: 29.4%; MCBs with additions: 31.7%) [24]. Further details on the clinical outcomes can be found in Table 4.

#### 3.4.3. Cost and Preparation Time

The cost and preparation time were only recorded in studies set in the hospital setting. Three studies out of the eight hospital-based studies reported on the costs involved in the use of MCBs and COM PN. Two reported that the total hospitalisation costs when using MCBs were less than when using COM PN; (MCB only: EUR 23,495.84, 95% CI 22,542 to 24,519; MCBs with additions: EUR 25,769.51, 95% CI 24,799 to 26,835; COMs: EUR 26,494.60, 95% CI 25,500 to 27,565) (converted from US dollars) [24]; (MCBs: EUR 7273.04; COM: EUR 9779.68) (converted from Chinese Yuan Renminbi) [30]. The same two studies reported that PN-related costs when using MCBs compared to COM were also less, apart from in the MCB with additions group (MCB only EUR 504.25, 95% CI 460 to 553; MCBs with additions: EUR 1042.48, 95% CI 955 to 1138; COMs: EUR 946.92, 95% CI 592 to 1032) (converted from US dollars) [24]; (MCBs: EUR 2147.03, COMs: EUR 7956.62) (converted from Chinese Yuan Renminbi) [30]. One study reported that the cost-effectiveness ratio of no infection and no re-operation was lower in MCBs [30], and the other stated that there were no differences between MCBs and COM PN in total cost, total nutrition support cost, PN drug cost, and EN drug cost; MCBs: EUR 6054.38, SD 2659; COM: EUR 6432.94, SD 2381; *p* = 0.216 (converted from Chinese Yuan Renminbi) [32]. Four out of the ten studies reported on PN preparation time, and all stated that the MCB preparation time was less than the COM [27,28,29,31]. Three studies did not report on any data on cost or preparation time [12,25,26]. Further details on cost and preparation time can be found in Table 5.

## 4. Discussion and Limitations of the Evidence and Review Process

This review investigated the available evidence for comparing the nutritional, clinical, and cost outcomes in patients prescribed PN in the format of MCBs or COM PN and found that MCBs may be more convenient and cost-effective than COM PN, at least in the hospital setting. Overall, in the hospital setting, MCBs provided adequate, well tolerated PN in a safe, timely, and cost-effective manner, and in some instances improved nutritional status to a greater extent than COM PN. The included hospital-based studies also demonstrated that compared to COM PN, MCBs showed low or no differences in post-operative infections and CRBSIs, metabolic disturbances, deaths, and length of hospital stay or readmissions. This is further substantiated by a study published after our literature search, which was a point prevalence study conducted with 263 patients, in the hospital setting, over two time periods and which compared the increased use of MCBs vs. compounded regimens. This study concluded that the wider use of in-hospital MCBs could lead to a reduced need for COM PN, resulting in cost savings whilst maintaining patient outcomes [34].

In the home setting, there was only one study reporting on nutritional adequacy, using the BMI [25,27]. Despite this agreeing with the hospital-based studies, no reliable conclusions can be drawn without further evidence, and there were no data recorded in relation to HPN preparation or cost. Both home-based studies agreed with the hospital-based studies and reported no difference between catheter-related bloodstream infections in MCBs compared to COMs. However, only one home-based study reported on metabolic complications and survival, and, as with nutrition adequacy, the data agreed with the hospital-based studies, but without further evidence no reliable conclusions can be drawn.

There were no reports given on patients’ experiences or QoL when receiving MCBs, highlighting a paucity of data in this area. A previous review looking at QoL in patients receiving PN at home concluded that QoL was lower in these patients compared to the general population, but the certainty of evidence was very low [35], indicating that further high-quality work is required in this area. Tools dedicated to assessing QoL in patients with PN have been previously developed [36] and investigated, with studies showing that QoL can be negatively affected by an increased frequency of PN infusions per week [37], the type of underlying disease [37,38], a shorter duration of PN [38], and living alone [38]. Further studies are required to ascertain the impact of specific PN infusion types, mainly MCBs, on QoL in both the home and hospital settings. Patients’ experiences of treatments received in acute settings are important, although it is worth noting that in some instances PN in hospitals was only received for short periods. In the treatment of chronic IF, QoL is a high priority, as patients live with HPN for longer periods over their life course.

As with QoL, relatively few studies reported on the cost implications. Those that did were hospital-based and reported either potential savings or no potential losses when using MCBs compared to COM PN in the hospital setting. However, the reported cost varied with regards to currency and which factors had been included in studies’ cost analysis, such as total hospitalisation costs, only PN-related costs such as the cost of labour in preparing individually compounded bags, or the occurrence and cost implications of infections or subsequent operations. Clearly, further work on the standardised cost implications of COMs vs. MCBs is required, both in the home and hospital setting.

The results of this systematic review are generally consistent with similar reviews and the previous review by Alfonso and colleagues [20]. Alfonso concluded that while the quantity and quality of studies is limited, the evidence suggests that MCBs in the hospital setting offer several advantages compared with COMs and multi-bottle systems, including fewer infections, shorter length of hospital stay, and reduced time and labour [20]. Another review considered the safety, cost, and clinical outcomes of the use of premixed PN, concluding that premixed is a definitive step towards the standardisation of PN therapy and that compared to COMs it may offer reduced bloodstream infections and a decreased cost [39].

This review was limited mainly by the low number of studies identified that prospectively measured the use of MCBs compared to COM PN, either in the hospital or home setting. Although three randomised controlled trials and one prospective study in hospitalised patients demonstrated higher-quality evidence supporting positive outcomes with MCBs, more than half of the included studies were conducted retrospectively and therefore subject to limitations, including selection bias. In this study, PN in hospitals and at home varied considerable in length; there were even major differences in PN given in acute settings which varied from hours to months.

Furthermore, most studies failed to report if patients were receiving any additional tube feeding or oral intake, which may have impacted the outcomes. Another limitation was that only two studies were conducted in a home setting, with one of these providing very little relevant data. However, these two studies have provided early evidence that clinical outcomes in the home setting may not differ from the hospital setting. Moreover, as the heterogeneity between the studies was high, it was difficult to draw meaningful conclusions, and conducting a meta-analysis was not possible. One further limitation of the evidence was the variation and lack of international standardisation in the terminology used for different types of PN bags and for the varied clinical reasons that patients may require PN, limiting the ability to compare patient populations and PN exposure across studies.

## 5. Implications for Practice, Policy, and Future Research

The evidence from the studies included here and in the previous review by Alfonso and colleagues [20] does not raise any concerns relating to the safety and efficiency of using MCBs over COMs. However, more high-quality research in this area is required, specifically on QoL.

## Figures and Tables

**Figure 1 nutrients-16-03964-f001:**
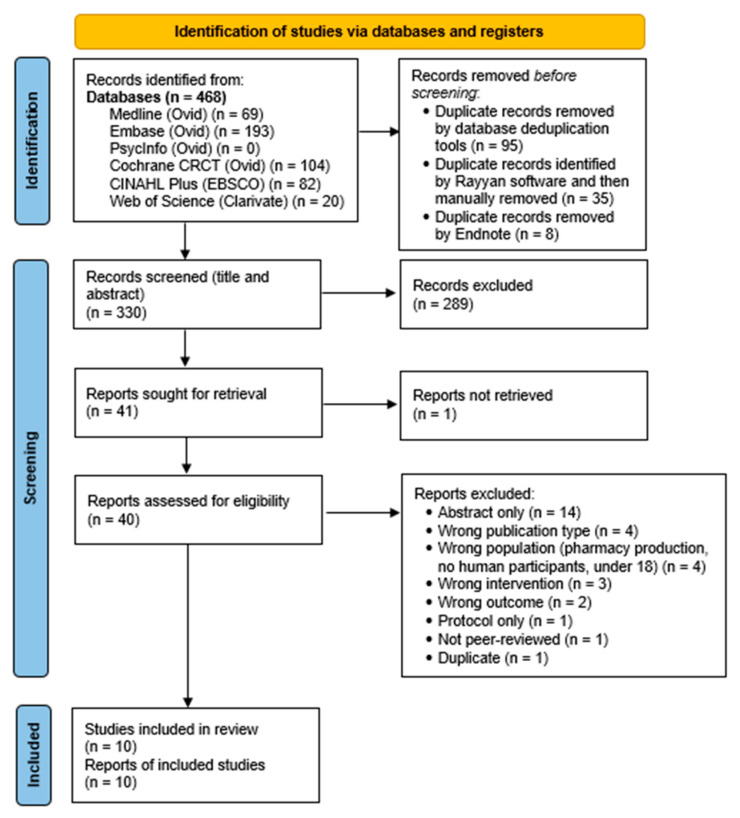
PRISMA 2020 flow diagram for a systematic review of the outcomes in patients receiving parenteral nutrition as multi-chamber bags [33].

**Table 1 nutrients-16-03964-t001:** Study and participant characteristics of the included studies.

	Author, Year (Country) Language	Study Design and Setting	Sample Size (*n*)	Format of PN Administration *n* (%)	Female *n* (%)	Mean Age (SD) Years	Ethnicity *n* (%)
Hospital	Banko, 2019 [24] (USA) English	Retrospective cross-sectional in hospital	84,564	MCB: 17,892 (21.2) COM: 66,672 (78.9)	47,246 (55.9)	Age range *n* (%): MCB: 18–44: 2376 (13.3) 45–64: 5439 (30.4) 65–74: 3986 (22.3) 75–84: 3829 (21.4) 85+: 2262 (12.6) COM: 18–44: 12,635 (19.0) 45–64: 22,710 (34.1) 65–74: 14,056 (21.1) 75–84: 11,858 (17.8) 85+: 5413 (8.1)	MCB: Hispanic 990 (5.5) NH White 7377 (41.2) NH Black 1324 (7.4) NH other 895 (5.0) Unknown 7306 (40.8) COM: Hispanic 2841 (4.3) NH White 15,868 (23.5) NH Black 2732 (4.1%) NH other 2909 (4.4) Unknown 42,532 (63.7)
	Goh, 2022 [26] (Singapore) English	Retrospective cross-sectional in hospital	172	MCB: 135 (78.5%) COM: 37 (21.5%)	MCB: 52 (38.5) COM: 16 (43.2)	MCB: 63 (15.41) COM: 66 (13.20)	NR
	Jia, 2015 [27] (China) English	Randomised controlled parallel group trial in hospital	458	MCB: 226 (49.3) COM: 232 (50.7)	MCB: 92 (40.7) COM: 87 (37.5)	MCB: 55.8 (13.1) COM: 56.3 (11.7)	MCB: Chinese Han 216 (96) Other Chinese 8 (3.5) Other 2 (0.5) COM: Chinese Han 220 (94.8) Other Chinese 11 (4.7) Other 1 (0.4)
	Park, 2020 [28] (Korea) English	Retrospective cross-sectional in hospital	1493	Commercial: 1443 (96.5) 86.2% = MCB COM: 50 (3.5)	625 (43.4)	62.5 (15.4)	NR
	Xi, 2021 [29] (China) Chinese	Randomised controlled trial in hospital	235	MCB: 122 (51.9) COM: 118 (50.2)	MCB: 42 (35.0) All in one: 47 (40.9)	MCB: 58.96 (10.65) All in one: 57.07 (11.63)	NR
	Xie, 2018 [30] (China) Chinese	Retrospective cohort in hospital	249	MCB: 122 (49.0) COM: 127 (51.0)	MCB: 50 (41.0) COM: 50 (39.4)	MCB: 57.7 (12.7) COM: 57.3 (10.3)	NR
	Yu, 2017 [31] (China) English	Randomised controlled trial in hospital	239	MCB: 121 (50.6) COM: 118 (49.4)	MCB: 54 (45.0) COM: 56 (47.0)	MCB: 59.83 (12.42) COM: 59.75 (12.56)	NR
	Zhao, 2018 [32] (China) English	Retrospective cohort in hospital	64	MCB: 30 (46.9) COM: 34 (53.1)	MCB: 5 (17.0) COM: 13 (38.0)	MCB: 59 (15.0) COM: 59 (10.0)	NR
Home	Crooks, 2022 [12] (UK) English	Retrospective cohort in the home	123	New grp: MCB 45 (42.9) COM 60 (57.1) Switch grp: 18 (100.0)	New grp: 76 (72.4) Switch grp: 8 (44.4)	New grp: MCB 53 (17.1) COM 59 (13.3) Switch grp: 55.6 (14.1)	NR
	Fernández- Argüeso [25] (Spain) 2024	Prospective cohort at home	130	MCB: 43 (33.1) COM: 87 (66.9)	MCB: 30 (69.0) COM: 57 (66.0)	MCB: 59 (13.0) COM: 58 (11.0)	NR

*n*: number, SD: standard deviation, PN: parenteral nutrition, MCB: multi-chamber bag, COM: compounded, NH: Non-Hispanic, yrs: years, NR: not reported, New grp: newly commenced HPN group, Switch grp: switching from customized to MCB group.

**Table 2 nutrients-16-03964-t002:** Exposure characteristics.

	Author, Year	Main Indication for PN *n* (%)	Nutritional Content of PN Bags	Tube Feeding/Oral Intake	Mean Duration of PN	Duration of Follow-Up
Hospital	Banko, 2019 [24]	Nutritional deficiency 84,564 (100.0)	NR	NR	*n* (%) MCB: 1–2 days, 3059 (17.1) 3–4 days, 4595 (25.7) 5–9 days, 6739 (37.7) 10+ days, 3499 (19.6) COM: 1–2 days 24,920 (37.4) 3–4 days 12,491 (18.7) 5–9 days 17,681 (26.5) 10+ days 11,580 (17.4)	No follow-up. Cross- sectional Outcomes, 30- and 90-days all-cause readmissions
	Goh, 2022 [26]	MCB: GI obstruction 46 (34.1) Post-op comp 58 (43.0) Severely maln 15 (11.1) GI haemorrhage 3 (2.2) Others 13 (9.6) COM: GI obstruction 8 (21.6) Post-op comp 18 (48.6) Severely maln 5 (13.5) Others 6 (16.2)	MCB: SmofKabiven 1206 mL, 800 kcal, 38 g Protein, 85 g glu, 34 g lipid In mmol: Na 30, K 23, Ca 1.9, Mg 3.8, PO4 9.9, Zn 0.03 COM: Individualised	NR	13–15 h between initiation of PN and subsequent blood test	13–15 h post initiation of PN on Day 1
	Jia, 2015 [27]	Underwent surgery: 397 (86.7) High-complexity surgery: 283 (62.0)	MCB: Oliclinomel N4 1.5 L with electrtolytes (80% olive oil, 20% soybean oil) COM: soybean oil-based lipid PN regimen using Intralipid	Nil from days 0–5. From day 6 onward, liquid oral or enteral nutrition could be added to the study treatment	MCB: 8.4 (SD 3.7) days COM: 8.2 (SD 3.7) days Max duration 14 days. Could be initiated 3 days pre-surgery	Study treatment of HPN was between 5 and 14 days
	Park, 2020 [28]	Neoplasms: 761 (53.0) Surgery: 410 (28.0) Organ systems Digestive: 186 (13.0) Respiratory: 87 (6.0) Circulatory: 74 (5.0) Blood/blood-forming organs/immune: 54 (4.0) Others: 277 (19.0)	Where the bags provided lipid, this could be either soybean, MCT, olive oil, or fish oil	NR	NR. The average hospital PN duration was 17.8 ± 52.6 days	N/A
	Xi, 2021 [29]	Elective moderate or major abdominal surgery	MCB: MCT/LCT fat emulsion/aa (16)/glu (36%) electrolyte injection COM: MCT/LCT fat emulsion 250 mL:50 g, compound aa injection (18AA-II) 250 mL:28.5 g, glu injection 100 mL:50 g, concentrated Cl_2_ NaCl injection 10 mL:1 g, KCl for injection 1 g, MgSO_4_ injection 10 mL:2.5 g, Ca gluconate injection was 10 mL:1 g, and sodium glycerophosphate injection 10 mL:2.16 g	Patients receiving “other nutritional therapy” concurrently were excluded	5 d starting on first day after the surgery and continued for 5 days	Treatment period 1–6 days after operation
	Xie, 2018 [30]	Gastric, colon cancer, benign gastric, colon tumour, GI tumours, rectal malignant tumours, duodenal secondary malignant tumour, and other GI surgery	MCB: fat emulsion, AA (17%), glucose 11% injection (Carvin) COM: glucose and MCT/LCT fat, AA, all vitamins/minerals and electrolytes	NR	NR	NR
	Yu, 2017 [31]	Elective open abdominal surgery for post-op PN	MCB: 1875 mL, 1435 kcal, 60 g AA, 120 g glucose, 37.5 g LCT, 37.5 g MCT COM: 1886.5 mL, 1439 kcal, 61 g AA, 120 g glucose, 75 g LCT, 0 g MCT	Oral and/or enteral nutrition covered up to 20% of daily caloric intake	Up to post-op day 6	Bloods monitored post-op days 1–7. Mortality checked at day 30 by phone call
	Zhao, 2018 [32]	Gastric cancer patients after gastrectomy	MCB: AA 60.38 ± 6.35; fat 63.97 ± 6.26; dextrose 200.51 ± 36.26 COM: AA 60.38 ± 6.35; fat 59.76 ± 5.11; dextrose 200.51 ± 36.26	NR	MCB 8 ± 3 days COM 7 ± 5 days	Followed up from the first day of post-op PN to discharge
Home	Crooks, 2022 [12]	New group All chronic IF MCB: Cancer 48 (80.0), Non- cancer 12 (20.0) COM: Cancer 11 (24.0), Non- cancer 34 (76.0) Switch grp: Cancer 2 (11), Non-cancer 16 (89)	NR	NR	New grp: MCB: 99 days COM: 170 days Switch grp: MCB: 269 days COM: 411 days	(Catheter days) New grp: MCB 5914 COM 7641 Switch grp: MCBs 4834 MCB 7401
	Fernández-Argüeso, 2024 [25]	Advanced cancer and intestinal occlusion or sub-occlusion	Mean (SD) MCB: energy 28.7(7.6) COM: energy 29.4 (7.9) glucose 3–6 g/kg/day AA1.0 g/kg/day Lipid < 1.0 g/kg/day/7–10 g/day EFA Vitamins/trace elements added where required	NR	Median days (IQR) MCB: 64 (126) COM: 64 (113)	2007 to 2022. Followed up every 2 weeks in the first 2 months, and every 1–3 months thereafter

*n*: number, PN: parenteral nutrition, MCB: multi-chamber bag, COM: compounded, New grp: newly commenced HPN group, Switch grp: switching from COM to MCB group, N/A: Not applicable, SD: standard deviation, EFA: essential fatty acid, IF: intestinal failure, MCT: medium-chain triglycerides, LCT: long-chain triglycerides, Comp: complications, energy expressed as Kcals/kg/day, AA: amino acid, maln: malnourished.

**Table 3 nutrients-16-03964-t003:** Nutrition, weight, and patient experience.

	Author, Year	Met Nutritional Requirements	Weight (kg)/BMI (kg/m^2^)	Patient Experience or Quality of Life
Hospital	Goh, 2022 [26]	Mean (95% CI) after 1 day of PN MCB: Calories, kcal/kg of body wt: 14.2 (13.8–14.7) protein, g/kg of body wt: 0.7 (0.6–0.7) COM: Calories, kcal/kg of body wt: 18.2 (17.0–19.5) protein, g/kg of body wt: 1.3 (1.2–1.3) No significant electrolyte derangements nor increased electrolyte replacement peripherally in the following day for both groups	Mean BMI (95% CI) MCB: 23.0 (22.3–23.7) COM: 20.6 (19.0–22.2); *p* = 0.0041	NR
	Jia, 2015 [27]	No difference in time to achieve adequate enteral intake (following 5 days of PN) (median time: olive 2.0 days versus soybean 2.0 days; log rank *p* = 0.786) Mean prealbumin (ITT) (mg/dL) (SD) MCB baseline 15.08 (4.64) *n* = 213 COM baseline 15.15 (5.01) *n* = 219 MCB day 5 15.66 (5.12), *n* = 217 COM day 5 13.95 (5.05), *n* = 218, LSGM ratio [95% CI] 1.12 [1.06, 1.19]; *p* = 0.0002 MCB day 14 EOT 17.24 (6.82) *n* = 210 COM day 14 EOT 15.15 (6.37) *n* = 219, LSGM ratio [95% CI] 1.16 [1.08, 1.24]; *p* = 0.001	Mean (SD) BMI MCB: 21.7 (3.9), *n* = 217 COM: 21.8 (3.9), *n* = 226; *p* = 0.667	NR
	Xi, 2021 [29]	Prealbumin trends comparable in both groups. Post-op nutritional index and retinol-binding protein not statistically different	NR	NR
	Yu, 2017 [31]	Prealbumin levels increased in MCB group by 2.70 ± 5.69 mg/dL and 2.59 ± 5.61 mg/dL in PPS and FAS (*p* < 0.001), respectively, while they remained stable in the control group (Prealb PPS = 0.36 ± 4.69 mg/dL, *p* = 0.465 and Prealb FAS = 0.29 ± 4.95 mg/dL, *p* = 0.606)	Mean Height: MCB 165.58 cm COM163.85 cm, *p* = 0.066 Weight: MCB 61.39 kg COM 60.8 kg, *p* = 0.557	NR
	Zhao, 2018 [32]	Mean nutritional risk screening 2002 (SD) at baseline MCB: 3.0 (1.0) COM: 4.0 (1.0)	Mean BMI (SD) Difference pre/post-TPN: MCB: −1.11 ± 1.06 COM: −0.90 ± 0.98 Difference between MCB and COM: 0.21, *p* = 0.416	NR
	Fernández-Argüeso, 2024 [25]	Mean (SD) MCB: Total cholesterol (mg/dL) 163.0 (47.0) CONUT index (score) 8.2 (2.8) MUST index (score) 1.9 (0.3) COM: Total cholesterol (mg/dL) 205.0 (205.0) CONUT index (score) 7.2 (2.4) MUST index (score) 1.9 (0.3) When nutritional data (body weight, BMI, and CONUT index both at baseline and at follow-up) were compared in those patients with and without CRBSIs, no significant results were obtained (*p* > 0.05 for all comparisons)	MCB: Mean (SD) weight 55.0 (14.0) Mean (SD) BMI 21.1 (5.5) COM: Mean (SD) weight 53.0 (13.0) Mean (SD) BMI 20.2 (4.4)	NR

BMI: Body mass index, NR: not reported, MCB: multi-chamber bag, COM: compounded, CI: confidence intervals, wt: weight, EOT: end of treatment, SD: standard deviation, CONUT: controlling for nutrition index. MUST: malnutrition universal screening tool, CRBSI: catheter-related bloodstream infection, ITT: intention to treat, PPS per protocol set, FAS: full analysis set, LSGM: least square geometric mean.

**Table 4 nutrients-16-03964-t004:** Clinical outcomes.

	Author, Year	Infections/Sepsis	Metabolic Disturbances	Mortality	Length of Stay	Readmission All Cause	PN Modification
Hospital	Banko, 2019 [24]	Mean BSI % risk (95% CI) MCB only: 2.1 (2.0–2.2) MCB adds: 7.0 (6.8–7.2) MCB overall: 5.5 (5.4–5.7) COM: 6.8 (6.7–6.9)	NR	NR	Mean LoS, days (95% CI) MCB: 12.4 (12.0–12.9) COM: 12.3 (11.9–12.7)	Mean 30 day % (95% CI) MCB only:18.5 (18.3–18.6) MCB adds: 22.0 (22.0–22.3) MCB total: 21.0 (20.9–21.1) COM: 20.5(20.4–20.6) Mean 90 day % (95% CI) MCB only: 26.8 (26.6–27.0) MCB adds: 31.7 (31.5–31.8) MCB total: 30.2 (30.1–30.3) COM: 29.4 (29.4–29.5)	NR
	Goh, 2022 [26]	NR	NR	NR	NR	NR	MCB was unmodified on the studies day (day 1). Both MCB and COM received top-up K, PO4, and Mg
	Jia, 2015 [27]	Number of patients with infections *n* (%): MCB 8.0 (3.6) COM 24 (10.4) *p* < 0.01	All biochemistry measures remained NORMAL Mild cholestasis was observed in all	Deaths (*n*) MCB 1 COM 4	Mean LoS, days (SD) MCB 16.92 (4.99) COM 18.1 (8.65) *p* = 0.7823	NR	NR
	Park, 2020 [28]	NR-MCB and COM not compared.	NR	NR-MCB and COM not compared	NR-MCB and COM not compared	NR-MCB and COM not compared	NR-MCB and COM not compared
	Xi, 2021 [29]	NR	Post-op inflammatory indicators in each group were not statistically different	NR	NR	NR	NR
	Xie, 2018 [30]	Post-op infections *n* (%) MCB 13 (10.6) COM 33 (26.0) *p* = 0.003	NR	No deaths in either group	Mean LoS, days: MCB 17.6 COM 18.5 *p* = 0.408	NR	NR
	Yu, 2017 [31]	NR	At post-op day 7, all bloods were comparable between groups	No differences at 30 days 30-day mortality rate in both groups was 0%.	No differences at 30 days	NR	Vitamins and trace elements were added to the infusion bag as required
	Zhao, 2018 [32]	NR	No significant difference except total bilirubin and direct bilirubin were significantly higher in the MCB group	NR	MCBs had shorter overall LoS (19 ± 12 days) versus COMs (24 ± 13 days)	NR	NR
Home	Crooks, 2022 [12]	CRBSIs per catheter days MCB: 0.51/1000 COM: 0.39/1000 Incident rate ratio 1.29 (95% CI 0.17–9.65) Subgroup MCB: 0.21/1000 COM: 0.27/1000 Incident rate ratio 1.31 (95% CI 0.12–14.30)	NR	NR	NR	NR	NR
	Fernández-Argüeso, 2024 [25]	CRBSIs per catheter days MCB: 0.21/1000 COM: 0.28/1000 *p* > 0.05	Severe metabolic complications n (%) MCB: 0 (0) COM: 1.0 (1.2)	Median days survival (95% CI) MCB: 88 (43–133) COM: 98 (49–147) (χ^2^ log rank test = 0.012, *p* = 0.913)	NR	NR	NR

PN: parenteral nutrition, BSI: bloodstream infection, MCB: multi-chamber bag, MCB adds: MCB with additions, COM: compounded, CI: confidence interval, NR: not reported, LoS: length of stay, CRBSI: catheter-related bloodstream infection, EOT: end of treatment, B/line: baseline, SD: standard deviation.

**Table 5 nutrients-16-03964-t005:** Production and time costs.

	Author, Year	Cost of PN	Preparation Time	Number of Modifications to a PN Prescription
Hospital	Banko, 2019 [24]	Mean PN-related cost (2015 US dollars) (95% CI): MCB only 549 (501–603) * MCB with additions 1135 (1040–1240) * MCB overall 997 (912–1088) * COM 1031 (945–1124) Mean total hospitalisation cost (2015 US dollars) (95% CI): MCB only 25,594 (24,540–26,692) * MCB with additions 28,072 (26,975–29,213) * MCB overall 27,479 (26,412–28,590) * COM 28,861 (27,759–30,007)	NR	NR
	Jia, 2015 [27]	NR	“The preparation time for study treatment was significantly less for MCB compared with COM on all days assessed (*p* < 0.001 for all values)”	NR
	Park, 2020 [28]	NR	The time spent for preparing the nutrient solution in the MCB group was significantly shorter than that in the COM group (*p* < 0.0001)	NR
	Xi, 2021 [29]	NR	MCB prep time was 13 min shorter than that of COM	NR
	Xie, 2018 [30]	PN cost (thousand yuan): MCB 1.7, COM 6.3, *p* < 0.000 Total hospital expenses (thousand yuan): MCB 63.3, COM 75.0, *p* = 0.158 Medicine cost (thousand yuan): MCB: 21.3, COM 30.0, *p* = 0.027 Total hospitalisation cost (yuan) MCB: 57,619, COM 77,476, *p* < 0.001 Cost-effectiveness ratio (yuan) Without re-operation: MCB 586, COM 793 Without infection MCB 645, COM 1047. *p* < 0.05 Post-op infection rate and the overall costs in the MCB group [(10.6%, (63.3 ± 28.4) × 103 RMB)] was decreased compared to COM group [(26.0%, (75.0 ± 28.8) × 103 RMB)], *p* < 0.05.	NR	NR
	Yu, 2017 [31]	NR	Mean prep time mins (SD) Post-op day 1: MCB: 4.9 (4.41) COM: 12.13 (5.62) *p* < 0.001 Post-op day 5: MCB: 4.56 (3.15) COM: 11.77 (4.79), *p* < 0.001	NR
	Zhao, 2018 [32]	Mean total cost (RMB) (SD) MCB: 47,961.31 (21,059.16) COM: 50,916.42 (18,857.46), *p* = 0.216	NR	NR

PN: parenteral nutrition, COM: compounded, CI: confidence intervals, mins: minutes, SD: standard deviation, NR: not reported, * *p* < 0.05 compared to COM.

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
