# Peer review of "Outcomes for Patients Receiving Multi-Chamber Bags for the Delivery of Parenteral Nutrition: A Systematic Review"

_nutrients, 2024, doi:10.3390/nu16223964_

Round 1
Reviewer 1 Report
Comments and Suggestions for Authors
The article "Outcomes for patients receiving multi-chamber bags for the delivery of parenteral nutrition: A systematic review" provides a well-organized and thorough analysis of clinical outcomes associated with the use of multi-chamber bags in parenteral nutrition. The structure is clear, and the relevance of this delivery method is effectively conveyed, underscoring its importance in clinical practice. The authors performed a careful selection of included studies and a rigorous evaluation of the data, which enhances the credibility of their findings.
From a formal standpoint, the article contains only a few minor grammatical errors that could be easily corrected to improve readability. However, the discussion section could be expanded to provide more in-depth analysis of the data. Currently, interpretations and clinical implications are somewhat limited, and a more detailed discussion of long-term effects and comparisons with other parenteral nutrition methods would have further enriched the analysis.
In summary, this article represents a significant and well-executed contribution to the literature on multi-chamber bags for parenteral nutrition, with some room for improvement in the discussion of findings.
Author Response
Comments 1: The article "Outcomes for patients receiving multi-chamber bags for the delivery of parenteral nutrition: A systematic review" provides a well-organized and thorough analysis of clinical outcomes associated with the use of multi-chamber bags in parenteral nutrition. The structure is clear, and the relevance of this delivery method is effectively conveyed, underscoring its importance in clinical practice. The authors performed a careful selection of included studies and a rigorous evaluation of the data, which enhances the credibility of their findings.
From a formal standpoint, the article contains only a few minor grammatical errors that could be easily corrected to improve readability. However, the discussion section could be expanded to provide more in-depth analysis of the data. Currently, interpretations and clinical implications are somewhat limited, and a more detailed discussion of long-term effects and comparisons with other parenteral nutrition methods would have further enriched the analysis.
In summary, this article represents a significant and well-executed contribution to the literature on multi-chamber bags for parenteral nutrition, with some room for improvement in the discussion of findings.
Response 1: Thank you for your kind and positive feedback. We have gone over the article and corrected any errors and have added detail to the discussion section.
Reviewer 2 Report
Comments and Suggestions for Authors
minor revision
This manuscript investigates and describes the outcomes of patients undergoing parenteral nutrition dialysis with multi-cavity bags in hospital and community Settings. Although, in this manuscript, the author's argument is comprehensive and well-worded. However, there are still some unreasonable descriptions of the structure and details of the article. After careful review, the following suggestions are offered
Minor comments:
1, It is recommended to add several different citations in the introduction section. Line 46-60.
2, It is recommended to beautify the chart format.
3, The title of "4.1. Limitations of the evidence and the review process" is too abrupt, and it is suggested to merge with "4. Discussion".
Author Response
Comments 1: It is recommended to add several different citations in the introduction section. Line 46-60.
Response 1: Thank you for your helpful comments. Several additional references have now been added to the introduction section.
Comments 2: It is recommended to beautify the chart format.
Response 2: Tables have been altered where possible for clarity and readability.
Comments 3: The title of "4.1. Limitations of the evidence and the review process" is too abrupt, and it is suggested to merge with "4. Discussion".
Response 3: Thank you for your suggestion these sections have now been merged.
Reviewer 3 Report
Comments and Suggestions for Authors
Dear author
The study is of excellent quality and essential for the field. The introduction is perfect and precise. The method is appropriate and correct.
It would be best if you organized the tables to be better quality.
Another question is, why did the study include only females? I think this point will need comment.
Author Response
Comments 1: It would be best if you organized the tables to be better quality.
Response 1: Tables have been altered where possible for clarity and readability.
Comments 2: Another question is, why did the study include only females? I think this point will need comment.
Response 2: Thank you for your question. The review considered both males and females. Table 1 displays the numbers and % of females only to keep table information a minimum. However, the number and % of males in each study can be worked out from the female data provided. Similarly, section 3.2.1 provides the numbers and % of females across all included studies, allowing for the total number and % of males to be established from this information and the total number of participants.
Reviewer 4 Report
Comments and Suggestions for Authors
Dear Authors,
This systematic review aimed to examine the evidence-base on clinical outcomes and/or quality of life in adults receiving parenteral nutrition as multi-chambe or individually compounded bags in hospital and community settings.
A systematic review is a type of research methodology that aims to gather all available empirical data in a certain topic, evaluate it critically, and draw conclusions that provide a summary of the findings. Conducting a systematic review of the literature is a complicated and time-consuming task, due to the requirement to apply methodological rigor. The description of the research question, method, results, discussion of limitations and conclusions presented by the Authors is clear. The Authors presented their review, and especially the part concerning the method, extremely precisely, which allows replicability on the basis of the description. The Authors showed knowledge of the principles of creating a systematic review, which allowed for more effective management of the process of creating this type of article.
The strong suits of the article are: a comprehensive analysis (the work includes an overview of a wide range of studies, which allows for a synthetic approach to the topic) and methodological reliability (application of PRISMA guidelines and assessment of methodological quality increase the credibility of the results). However, there is a certain risk of a restriction on the literature (focused on English-language publications), which may lead to the exclusion of significant research in other languages).
At the same time, the need to develop various scientific problems in the form of a systematic review should be stressed. Knowledge of systematic reviews helps to face meet the challenges of the currently dynamically developing approach to evidence-based learning, creating the opportunity to participate in international cooperation.
The presented article is a valuable contribution to the field of research related to parenteral nutrition. Despite some limitations, its article strengths make it a solid source of knowledge for professionals dealing with dietetics and public health.
Author Response
Comments 1: Dear Authors,
This systematic review aimed to examine the evidence-base on clinical outcomes and/or quality of life in adults receiving parenteral nutrition as multi-chambe or individually compounded bags in hospital and community settings.
A systematic review is a type of research methodology that aims to gather all available empirical data in a certain topic, evaluate it critically, and draw conclusions that provide a summary of the findings. Conducting a systematic review of the literature is a complicated and time-consuming task, due to the requirement to apply methodological rigor. The description of the research question, method, results, discussion of limitations and conclusions presented by the Authors is clear. The Authors presented their review, and especially the part concerning the method, extremely precisely, which allows replicability on the basis of the description. The Authors showed knowledge of the principles of creating a systematic review, which allowed for more effective management of the process of creating this type of article.
The strong suits of the article are: a comprehensive analysis (the work includes an overview of a wide range of studies, which allows for a synthetic approach to the topic) and methodological reliability (application of PRISMA guidelines and assessment of methodological quality increase the credibility of the results). However, there is a certain risk of a restriction on the literature (focused on English-language publications), which may lead to the exclusion of significant research in other languages).
At the same time, the need to develop various scientific problems in the form of a systematic review should be stressed. Knowledge of systematic reviews helps to face meet the challenges of the currently dynamically developing approach to evidence-based learning, creating the opportunity to participate in international cooperation.
The presented article is a valuable contribution to the field of research related to parenteral nutrition. Despite some limitations, its article strengths make it a solid source of knowledge for professionals dealing with dietetics and public health.
Response 1: Thank you for your kind and positive feedback. We note your comment on exclusions of other non-English languages. However, we did not exclude based on publication language and did include and translate 2 studies published in Chinese language.